# Clinical Outcomes and Patterns of Traumatic Injuries Associated with Subway Incidents at a Level 1 Trauma Center

**DOI:** 10.3390/life15010051

**Published:** 2025-01-03

**Authors:** Bharti Sharma, Aubrey May B. Agcon, George Agriantonis, Sittha Cheerasarn, Navin D. Bhatia, Zahra Shafaee, Jennifer Whittington, Kate Twelker

**Affiliations:** 1Department of Surgery, Elmhurst Hospital Center, NYC Health + Hospitals/Elmhurst, 79-01 Broadway, Queens, NY 11373, USA; bokoa1@nychhc.org (A.M.B.A.); agriantg@nychhc.org (G.A.); cheerass@nychhc.org (S.C.); bhatian1@nychhc.org (N.D.B.); shafaeez1@nychhc.org (Z.S.); harrisj20@nychhc.org (J.W.); twelkerk1@nychhc.org (K.T.); 2Department of Surgery, Icahn School of Medicine at Mount Sinai Hospital, New York, NY 10029, USA

**Keywords:** trauma, severity, injury pattern, traumatic injury, subway, injury description

## Abstract

Objectives: Subway-related accidents have risen with advancements in the system. We aim to study the injury patterns from these incidents. Methods: This is a retrospective study from a single center, covering patients from 1 January 2016 to 31 December 2023. Patients were identified using International Classification of Diseases (ICD) injury descriptions and Abbreviated Injury Scale (AIS) body regions. Results: Out of 360 patients (total), 23.5% presented with head injuries with an AIS score ≥ 3. Patients with blunt trauma (93.99%) were in higher numbers than penetrating (5.74%) and burn trauma (0.3%). Overall, the mean Injury Severity Score (ISS) was 10.69, suggesting a broad range of traumatic injuries. ISSs for severe injuries (17–24) comprised 9.2%, moderate injuries (10–16) comprised 17.5%, and minor injuries (1–9) comprised 60.8%. Falls had the highest percentage of traumatic brain injuries (TBI) (65.60%) and fractures (67.50%). Assaults showed a significant occurrence of traumatic thoracic injuries (28.90%). Suicide attempts demonstrated a high percentage of traumatic amputations (30.80%). In the emergency department (ED), most patients (69.4%) were admitted for further care, such as trauma, neurosurgery, or other care. Of these patients, 0.5% died in the ED, 0.5% died on arrival, and 1.04% died within 15 min of arrival. The mortality rate among serious fall patients was 17.20% compared to the suicide and train-struck groups at 37.90% each. Conclusions: There were high occurrences of TBIs, and fractures, thoracic injuries, and amputations. Numbers of patients with blunt trauma were a lot higher than those with penetrating and burn trauma. The mortality rates observed in the suicide and train-struck groups were higher than those in patients with severe falls.

## 1. Introduction

Subway systems play a vital role in urban transportation across the United States, supporting the daily commutes of millions. However, subway-related injuries (SRIs) present a significant public health concern, especially in densely populated cities like New York City (NYC), where the subway system transported 2,027,286,000 passengers in 2023 [1]. From 1990 to 2003, NYC recorded over 668 fatalities related to subway incidents, with numerous injuries leading to severe trauma, such as extremity fractures and amputations [2,3,4]. As subways continue to serve as a vital mode of transportation for millions, accidents within these systems result in a wide variety of traumatic injuries that demand specialized medical attention. The complexity of these injuries, ranging from severe head trauma to intricate fractures, often requires coordinated efforts across multiple clinical disciplines, especially in high-level trauma centers [4].

Injury patterns commonly observed in subway accidents often include traumatic brain injuries (TBI), such as subarachnoid hemorrhages (SAHs), epidural hemorrhages (EDHs), and subdural hemorrhages (SDHs), all of which carry the potential for long-term neurological deficits [5]. Fractures of facial bones, ribs, and the skull base also occur frequently, further complicating patient outcomes [6]. Additionally, soft tissue injuries such as lacerations of the scalp and lung contusions, alongside serious conditions like traumatic pneumothoraxes, highlight the broad scope of trauma experienced by subway accident victims [7]. These patterns emphasize the critical role of trauma centers in not only treating these injuries but also in collecting and analyzing data to inform future prevention and treatment strategies.

Research indicates that the most common isolated severe injury was in the lower extremity, and the most common combinations of severe injuries were in the head and lower extremity, and the head and thorax [8,9]. Furthermore, Lin and Gill examined subway-related fatalities in NYC between the years of 2003 and 2007. They found rates of head injury, torso injury, and amputation of 88%, 76%, and 33%, respectively [10]. A study conducted at Bellevue Hospital in New York revealed that from a patient pool of 254, 17 patients were seen due to suicide, 3 were seen due to assault, 49 were hit by a train, and 44 fell onto the tracks [11]. This study expanded on the types and locations of injuries.

Despite the growing recognition of subway-related trauma, the body of research on clinical outcomes and injury patterns remains sparse. Studies focusing on the mechanisms and consequences of such accidents are crucial for improving patient care, enhancing trauma system readiness, and guiding public health interventions aimed at reducing the incidence and severity of these events. We hypothesize that subway injuries may lead to various traumatic injury patterns along with poor clinical outcomes and our objective is to study various traumatic injury patterns associated with subway traumatic incidents and related clinical outcomes.

There is a limited amount of published data from the United States regarding subway injuries [11,12]. Hence, our primary objective is to study various traumatic injury patterns and mechanisms of injury associated with subway traumatic incidents, whereas our secondary objective is to study various clinical outcomes/discharge dispositions associated with these injury patterns, for example, outcomes of Discharge to a Home/Shelter, hospice, Inpatient Psychiatry Care, Left Against Medical Advice, Met brain death criteria, mortality, Other Acute Care Hospital Emergency Department (ED), or Police Custody/Jail/Prison, Skilled Nursing Facility, and Sub Acute Rehabilitation/Inpatient Rehabilitation. By delving into the specific injury patterns seen in subway-related incidents, particularly within the context of a level 1 trauma center, this study seeks to expand the current understanding of these complex cases and that of the development of more effective clinical outcomes.

## 2. Methods

This is a single-center, retrospective review conducted at a level 1 trauma center verified by the American College of Surgeons in Queens, New York City. We included all patients who presented with traumatic SRIs between 1 January 2016 and 31 December 2023. All patients with subway-related injuries were included. We excluded patients with no subway-related injuries.

Patient data were requested from the trauma registry at our facility (Elmhurst Hospital Center). Our center’s trauma registry utilizes NTRACS software (American College of Surgeons National Trauma Registry System, https://www.facs.org/quality-programs/trauma/quality/national-trauma-data-bank/). Patients were identified based on the injury mechanism, cause of injury, primary mechanisms (lCD9 or lCDL0 E-Code), and the Abbreviated Injury Severity (AIS) score. The AIS score ranges from 1 to 6 per body region.

Utilizing trauma registry data and adhering to inclusion and exclusion criteria, we identified a final patient group of 360 individuals. The medical charts of the patients were reviewed, and all relevant information required for this study was collected.

We collected data using a data collection tool (Excel sheet or spreadsheet). We incorporated all data elements into this tool. Examples of data elements are demographics (for example: age, sex, race, ethnicity), AIS, GCS, ISS, injury pattern, discharge disposition, mortality status, mechanism of injury (MOI), and others. The dataset underwent several preprocessing steps to ensure data integrity, confidentiality, and suitability for statistical analysis. Unique identifiers, including medical record numbers (MRNs), dates of birth (DOBs), and patient names, were removed to deidentify the dataset. This process was carried out to maintain patient privacy in compliance with ethical research standards.

The AIS regions for patients were evaluated, with regions of interest being the head, face, neck, thorax, abdomen, spine, upper extremity, lower extremity, and external injuries. Furthermore, the following mechanisms of injury were studied: assaults, falls, suicide, and train strikes. The types of subway-related trauma were analyzed under the mechanisms of injury. The types of subway-related trauma include the sum of facial fractures, vertebral fractures, SAHs, SDHs, skull injuries, solid organ injury, traumatic thoracic injuries, EDHs, hand/foot fractures, Abdominal Injury, and diffuse TBI. Also, gender, race, age, and ethnicity data were collected under the mechanisms of injury.

Furthermore, the clinical outcomes were collected under the mechanisms of injury. The clinical outcomes include Discharge to a Home/Shelter, hospice, Inpatient Psychiatry Care, Left Against Medical Advice, Met brain death criteria, mortality, Other Acute Care Hospital ED, or Police Custody/Jail/Prison, Skilled Nursing Facility, and Sub Acute Rehabilitation/Inpatient Rehabilitation.

For Statistical analysis, we used the Phi coefficient correlation matrix to examine relationships between trauma injury patterns associated with subway trauma and clinical outcomes based on discharge disposition. Injury trends and discharge outcomes, such as discharge to a skilled nursing facility, home, subacute rehabilitation, and hospice, were abstracted for the electronic medical record.

First, we evaluated connections using the Chi-square test. However, this approach was limited by assumptions that were not met, such as the mutual exclusion of some discharge dispositions and low predicted frequencies in some cells. We recorded all variables as binary indicators to better capture associations and computed Phi coefficients for every pair. A heatmap displayed the results, highlighting important connections between injury trends and discharge outcomes that can guide post-discharge care and resource allocation. Further descriptions of these statistical approaches are elaborated within the result section to avoid duplication.

The relationships between the MOIs and medical procedures were analyzed using the Chi-square test, a suitable method for evaluating associations between categorical variables like the mechanism of injury and binary medical procedure outcomes. The test identifies whether the observed distribution deviates significantly from chance.

The Revised Trauma Score (RTS), which incorporates the Glasgow coma scale (GCS), systolic blood pressure (SBP), and respiratory rate (RR), was calculated for each patient. For calculation, we used the following formula: RTS = 0.9368 GCS + 0.7326 SBP + 0.2908 RR. RTS was coded into the dataset as a composite score representing the physiological severity of trauma. By integrating the Revised Trauma Score (RTS) calculation, the mean imputation for missing data, and Chi-square tests for categorical variables, this study comprehensively examines subway-related trauma patterns and their associated clinical outcomes. This score was used as a predictor of clinical outcomes and included in subsequent statistical analyses.

## 3. Results

The occurrence of injuries by body region with an AIS score of three or greater, which is considered a severe traumatic injury per trauma patient, was evaluated. Table 1 represents common combinations of severe injuries per body region. The head region is the highest, followed by thorax region injuries and lower extremity injuries.

Table 1 of this paper presents the distribution of trauma types (blunt, burn, and penetrating) among subway-related injury cases. A total of 360 cases were included. Blunt trauma accounted for the majority of cases (93.9%), while penetrating trauma represented 5.8%, and burn injuries made up 0.3%.

The distribution of the ISS provides insights into the frequency of injuries based on their severity levels within the study population. Overall, the mean ISS of 10.69, with a standard deviation of 11.33, suggests a broad range of traumatic injuries. Most patients suffered from minor injuries, the ISS of which is between 1 and 9. In total, these patients constitute 60.8% (219 cases). Meanwhile, around 17.5% of cases (63 cases) were observed to have moderate injuries, with an ISS between 10 and 16. Severe injuries (with an ISS of 17–24) comprised 9.2% of the study cohort (33 cases).

Table 2 of this paper presents the demographic breakdown of subway-related trauma cases based on different mechanisms of injury (MOIs), including falls, assaults, train strikes, suicides, and other causes (such as electric injuries, strikes against objects, and other unspecified injuries). Age is reported as the mean and standard deviation, while gender, race, and ethnicity are displayed as counts and percentages. The data show that individuals involved in falls had an average age of 52.51 years, the oldest among the groups, while those in the “other” injury category had the youngest average age (33.62 years). Males were the predominant gender in all MOI categories, with the highest percentage in the assault category (92.60%). Regarding race, the majority of patients across most MOI groups were classified as “Other”, with the highest proportion observed in the assault category (74.10%). The ethnicity data reveal that a Hispanic origin was most common in the assault group (64.80%). This demographic information helps to contextualize the characteristics of patients based on their injury mechanisms.

Table 3 of this paper outlines the injury patterns associated with various MOIs in subway-related trauma incidents, specifically focusing on assaults, falls, other causes (such as electric injuries and being struck against objects), suicides, and train strikes. The table provides counts and percentages for different types of injuries, including brain-specific injuries, fractures, solid organ injuries, traumatic thoracic injuries, abdominal injuries, and traumatic amputations.

Notably, falls accounted for the highest percentage of brain-specific injuries (65.60%) and fractures (67.50%), indicating their prevalence in this MOI. In our patient population, we found fractures of the long bone, hand, foot, and face. Assaults showed a significant occurrence of traumatic thoracic injuries, with 28.90% of cases reporting these injuries. Falls accounted for the highest percentage of solid organ injuries (65.60%). Suicide attempts demonstrated a high percentage of traumatic amputations (30.80%), indicating a unique pattern in injury presentation. Overall, these data highlight the diverse injury patterns associated with different MOIs, emphasizing the need for tailored clinical approaches for each category.

Table 4 of this paper summarizes the clinical outcomes for trauma patients categorized by their MOI, including assaults, falls, other injuries (such as electrical or blunt force), suicides, and train-related incidents. It presents the frequency and percentage of various dispositions, illustrating the diverse trajectories patients experience following trauma.

The data reveal significant variations in outcomes across different injury mechanisms. For instance, a substantial percentage of patients from fall incidents (72.50%) were discharged to a home or to shelters, while only 1.30% of patients from assault and suicide incidents experienced similar outcomes. The mortality rate was notably high among suicide victims (37.90%), indicating the severity of injuries in this category. Other clinical outcomes include transfers to inpatient psychiatric care, skilled nursing facilities, and rehabilitation services, reflecting the multifaceted needs of trauma patients. This table highlights the critical role of targeted interventions and the necessity for tailored care pathways based on the MOI.

Table 5 of this paper presents the distribution of ED dispositions among patients involved in trauma cases. It details the frequency and percentage of various outcomes, including admissions, fatalities in the ED, deaths on arrival, deaths within 15 min of arrival, routine discharges to a home or self-care, and transfers to other hospital EDs.

The majority of patients (69.4%) were admitted for further care, while a smaller percentage experienced a fatal outcome, with 0.5% dying in the ED, 0.5% dying on arrival, and 1.04% dying within 15 min of arrival. Routine discharges accounted for 18.54% of the cases, indicating that a significant number of patients were stabilized and released. The data underscores the critical role of the ED in managing trauma cases, highlighting both the successful admissions and the challenges faced in life-threatening situations.

The RTS is a validated tool that assesses physiological derangements in trauma patients and predicts survival outcomes. Table 6 shows the analysis evaluated differences in Revised Trauma Scores (RTSs) across five categories of mechanisms of injury (MOIs): Assault, Fall, Other, Suicide, and Train-struck. It incorporates key indicators such as vital signs and Glasgow Coma Scale scores, making it a critical metric for trauma assessment and management. A one-way analysis of variance (ANOVA) revealed a statistically significant effect of MOI on RTS (F(4, N) = 28.70, *p* < 0.001 F(4, N) = 28.70, *p* < 0.001 F(4, N) = 28.70, *p* < 0.001), indicating that trauma severity varies significantly across the different mechanisms of injury. To further explore these differences, pairwise comparisons were conducted using Tukey’s Honest Significant Difference (HSD) test. Tukey’s HSD applies adjusted *p*-values (*p*-adj) to control the family-wise error rate, mitigating the risk of Type I errors that increase with multiple comparisons. This ensures the reliability and robustness of statistical inferences.

Post hoc analysis identified significant differences between groups. RTSs for Assault were significantly lower than those for Other, Suicide, and Train-struck (*p*-adj < 0.001 *p*-adj < 0.001 *p*-adj < 0.001), indicating greater trauma severity associated with assaults. Similarly, RTSs for Fall were significantly lower compared to those of Other, Suicide, and Train-struck (*p*-adj < 0.001 *p*-adj < 0.001 *p*-adj < 0.001). However, no significant differences were observed between Other and Suicide (*p*-adj = 0.991 *p*-adj = 0.991 *p*-adj = 0.991), Other and Train-struck (*p*-adj = 0.871 *p*-adj = 0.871 *p*-adj = 0.871), or Suicide and Train-struck (*p*-adj = 0.936 *p*-adj = 0.936 *p*-adj = 0.936). These findings suggest that while some mechanisms, such as assaults and falls, are associated with higher trauma severity, others (e.g., Other, Suicide, and Train-struck) may result in comparable physiological impacts as measured by RTS.

The adjusted *p*-values in Tukey’s HSD analysis were critical to ensuring the validity of pairwise comparisons. By accounting for the increased likelihood of false positives due to multiple comparisons, the adjusted *p*-values provide a rigorous framework for identifying true differences between groups. This approach reinforces the statistical integrity of the findings and ensures that significant results are not merely artifacts of chance.

Clinically, the results highlight the heightened trauma severity associated with mechanisms like assaults and falls, underscoring the need for targeted interventions during triage and resource allocation. Conversely, the comparable RTSs among Other, Suicide, and Train-struck suggest that these mechanisms may require similar clinical approaches. These findings affirm the utility of the RTS as a critical tool for trauma care and provide valuable insights for optimizing outcomes based on the mechanism of injury. The use of adjusted *p*-values further enhances the reliability of these conclusions, making them robust for both clinical and research applications.

Figure 1 of this paper represents the Phi coefficient correlation matrix for injury patterns and hospital disposition in subway-related trauma. The Phi coefficient correlation matrix provides valuable insights into associations between various binary clinical outcomes, informing patient care planning and resource allocation. Although, discharge to a home and discharge to subacute rehab are theoretically mutually exclusive options. Trauma patients were usually discharged to either one or the other, not both. A high correlation is observed between these two variables. Positive correlations were found between thoracic injury patterns and solid organ injuries, with a coefficient of 0.44, suggesting that patients with thoracic injuries are moderately likely to have associated solid organ injuries. This association aligns with clinical expectations, where thoracic injuries often involve solid organs.

Additionally, other meaningful associations were observed. There was a moderate positive correlation between amputation and discharge to hospice care (0.27), indicating that patients who undergo amputation may often be dealing with severe or end-of-life conditions and poor prognosis, underscoring the importance of hospice care planning for these patients. Negative associations were also notable, particularly between discharge to a home and mortality (−0.42). These associations imply that patients discharged to a home are generally in more stable conditions. Many other variable pairs showed weak or minimal correlations, such as solid organ injury patterns and psychiatric care discharge disposition outcomes, indicating that these factors may operate independently in this dataset.

Overall, the correlation matrix highlights expected clinical patterns, such as the relationship between discharge destinations and patient stability, while also revealing the importance of how data are coded, especially for mutually exclusive outcomes like discharges to a home and rehab. This correlation analysis allows clinicians to better anticipate patient needs post-discharge and make informed decisions about resource allocation.

The relationships between the MOIs and various procedures were analyzed using the Chi-square test, a suitable method for evaluating associations between categorical variables like the mechanism of injury and binary medical procedure outcomes. The test identifies whether the observed distribution deviates significantly from chance. Significant associations were found for Neurosurgery (NSGY) Intervention *p* = 0.0199 and Intubation *p* < 0.00001, indicating their likelihood is related to the mechanism of injury. No significant associations were found for Orthopedic Procedure, *p* = 0.266, Transfusion, *p* = 0.926, Mechanical Ventilation Days, *p* = 0.957, Vascular Access Insertion, *p* = 0.889, and Drainage Intervention, *p* = 0.563. The findings depend on meeting Chi-square assumptions, including adequate sample sizes and expected cell frequencies of at least five. While sparse data may affect reliability, the Chi-square test remains a robust tool for analyzing categorical relationships. These results highlight the importance of NSGY Intervention and intubation in trauma care, warranting further study to confirm these associations and explore their implications.

Table 7 illustrates the relationships between the mechanism of injury (MOI) and various procedures which were analyzed using the Chi-square test, a suitable method for evaluating associations between categorical variables like the mechanisms of injury and binary medical procedure outcomes. The test identifies whether the observed distribution deviates significantly from chance.

Significant associations were found for NSGY Intervention *p* = 0.0199 and Intubation *p* < 0.00001, indicating their likelihood is related to the mechanism of injury. No significant associations were found for Orthopedic Procedure, *p* = 0.266, Transfusion, *p* = 0.926, Mechanical Ventilation Days, *p* = 0.957, Vascular Access Insertion, *p* = 0.889, and Drainage Intervention, *p* = 0.563.

The findings depend on meeting Chi-square assumptions, including adequate sample sizes and expected cell frequencies of at least five. While sparse data may affect reliability, the Chi-square test remains a robust tool for analyzing categorical relationships. These results highlight the importance of NSGY Intervention and intubation in trauma care, warranting further study to confirm these associations and explore their implications.

## 4. Discussion

This study provides valuable insights into the injury patterns and clinical outcomes resulting from subway-related accidents, a critical area of research given the vast number of passengers utilizing urban transportation systems daily. With the ongoing advancements in subway infrastructure, the incidence of SRIs has notably increased, making it essential to understand the mechanisms, demographics, and clinical outcomes involved in these injuries.

Our findings indicate a significant prevalence of traumatic head injuries, specifically SAHs, EDHs, and SDHs, which accounted for a cumulative total of 78 cases among the 360 patients studied. Head injuries are particularly concerning, as they often result in long-term neurological deficits and complications.

Furthermore, the percentage of patients presenting with an Abbreviated Injury Scale (AIS) score ≥ 3 in the head region (23%) does not reflect the percentage of patients with severe head injuries (12%) from a 2020 Canadian Study [12]. The percentage of patients presenting with an AIS score ≥ 3 in the thorax region (16%) is also reflective of the percentage of patients with severe thorax injuries (8%) from the same 2020 study. Conversely, both studies report a similar prevalence of severe lower extremity injuries at 14%, underscoring a shared concern regarding these types of injuries across different populations. These findings underscore the need for a nuanced understanding of injury severity assessments and their implications for clinical outcomes and management strategies in trauma care.

Moreover, the study reveals that blunt trauma constituted the majority of cases (93.99%), with only 5.74% categorized as penetrating trauma. This pattern is consistent with previous research indicating that most SRIs arise from blunt forces associated with falls, collisions, or being struck by a train. In a 2009 study, a majority of cases were classified as blunt trauma (98%) [11,12]. The findings of a median ISS of 8, with an interquartile range of 10, indicate a broad range of injury severity, further reinforcing the necessity for trauma centers to be equipped to handle complex cases.

Our study reveals significant differences in the mechanisms and types of injuries sustained across various contexts: assaults, falls, suicide, train strikes, and other injury types. Notably, brain-specific injuries were most prevalent among fall cases, comprising 65.60% of all instances. This contrasts sharply with assault cases, where only 4.20% of injuries were brain-specific. Brain-specific injuries include SAHs, EDHs, SDHs, Diffused TBIs, and other intracranial injuries. Comparing this to the Bellevue Hospital study, they classified inter-cranial hemorrhage injuries in 27% of patients [11]. Even though the study does not specify the MOIs, it can be inferred that most of these data are from falling and train strikes because 49% and 44% of the patient population were classified as having been hit by a train and having fallen onto tracks, respectively. Furthermore, our study exhibits that fractures were predominantly associated with falls (67.50%), aligning with the Bellevue Hospital study data that indicates a 49% incidence of at least one fracture among patients [11]. To elaborate, both studies show that the most prevalent types of fractures are facial and vertebral fractures. Conversely, solid organ injuries were primarily observed in fall cases (65.70%) and somewhat in assault cases (8.60%). This distribution mirrors the Bellevue Hospital study’s indication of a 17% solid organ injury rate, highlighting the necessity for further investigation into the underlying causes and mechanisms that contribute to such injuries in different contexts [11]. Collectively, these findings underscore the importance of understanding the relationship between injury mechanisms and clinical outcomes to inform effective prevention strategies.

When comparing our study’s demographics with a 2009 study’s data, there are some key differences in age, gender, race, and ethnicity [10]. The 2009 study does not specify the MOIs. The mean age for falls in our study is higher (52.51 years) compared to other injuries like assaults (37.89 years), while the 2009 study does not provide age specifics. Racial and ethnic patterns differ notably. In our study, a majority of patients who experienced falls and assaults belong to “Other” races (50.40% and 74.10%, respectively), whereas Caucasians dominate in the 2009 study (32%) [10]. Hispanic patients are more prevalent in assault cases in our study (64.80%) compared to the 2009 study (28%) [10]. These differences likely reflect regional variations and suggest the importance of tailoring injury prevention strategies to local demographics.

In our study, the frequency of traumatic amputations varies across injury mechanisms, with falls and train strikes accounting for the highest percentages of amputations (23.10% and 30.80%, respectively). Assaults, suicides, and other injury types contribute smaller portions, with 7.70% each. The majority of these amputations are lower extremity amputations.

ANOVA revealed a statistically significant effect of the MOI on the RTS, indicating that trauma severity varies significantly across the different MOIs. We found significant associations for NSGY Intervention and intubation, indicating that their likelihood is related to the mechanism of injury. No significant associations were found for Orthopedic Procedures, Mechanical Ventilation Days, Vascular Access Insertion, and Drainage Intervention.

According to Maclean et al., their study presents more detailed classifications of amputation types, with the majority being lower extremity amputations, particularly below the knee (11 unilateral cases and 3 bilateral) [12]. Upper extremity amputations and mixed upper and lower extremity amputations are less common [13]. There are also a few cases of conversion from below-knee to above-knee amputations, highlighting the progression of injury severity in some patients.

While both datasets emphasize the prevalence of lower extremity amputations, the Maclean study provides more specificity regarding the level and type of amputation. Our study’s broad categorization does not capture the nuanced patterns of injury severity and progression reflected in the comparison study’s data.

The clinical outcomes in our study highlight the severity and complexity of trauma care. Among the 360 patients, a significant number required surgical interventions, such as craniotomies and orthopedic surgeries, emphasizing the severity of the injuries sustained. Despite this, the overall mortality rate remained relatively low, particularly among fall patients (17.20% mortality) compared to the higher rates observed in the suicide and train-struck groups (37.90% each). This suggests that timely and effective trauma care likely contributed to the favorable outcomes observed in many patients, with a large proportion of fall patients (72.5%) being discharged to a home or to a shelter.

In contrast, an 1987 London Underground study provided less detail on surgical interventions but showed several patients admitted to intensive care, neurosurgical wards, and psychiatric units, with some dying within hours of their injuries [13]. The 1987 study, while lacking percentage breakdowns, highlights the immediate critical outcomes: 23 patients were admitted to psychiatric units, and 14 were admitted to general hospital wards [13]. Deaths were relatively rare but significant, with four patients dying in the accident and ED within 4 h, and others dying in various units like the operating theater and ICU [13].

Paramedics play an essential role in patient care. The quality of care provided by medical teams, and paramedical/EMTs can be affected by the patient’s critical condition or the severity of injury. For example, prehospital intubation can result in higher mortality than in patients who are non-prehospital intubated. Interventions such as intubation might result in deleterious outcomes [14].

Additionally, a more recent study reviewed subway-related injury cases, revealing a mortality rate of 9.6% and that 25% of patients suffered extremity amputations, highlighting the severe nature of these injuries [15,16,17]. Of the survivors, 35.4% required transfer to psychiatric or rehabilitation services, with the remaining 53.4% were discharged, aligning with the trends observed in our study regarding the need for extensive rehabilitation and psychiatric care [14]. These data further emphasize the importance of addressing long-term outcomes in subway-related trauma, including psychological and rehabilitative care, as many survivors face ongoing physical and cognitive challenges.

Gender trends are consistent across multiple studies, with males dominating in traumatic injuries—74.40% for falls and 92.60% for assaults in our study, similar to the 2009 study’s 83% male rate. However, the 2009 study reports a higher percentage of female patients overall (52.6%), particularly compared to the 25.60% for falls in our study [10,18,19,20]. This recurring gender pattern is not only associated with subway injuries but is also with associated with road injuries [21,22,23], stab wounds [23,24,25], and gunshots [24].

Comparing the two, our study shows a broader range of clinical outcomes, including long-term care needs like rehabilitation, which were less detailed in the 1987 study. Many survivors, particularly those in our cohort who were discharged to rehabilitation or psychiatric care, may face ongoing challenges with physical and cognitive functioning. This echoes the broader need for attention to long-term outcomes, which remain underexplored in both our data and the comparison study.

Falls from heights are associated with high-energy impacts and are a major cause of morbidity and mortality. Although the reliability of height as an indicator of severity has been questioned, it remains one of the most commonly used parameters in trauma assessments. Research indicates a direct relationship between the height of the fall and the likelihood of death [26,27]. In SRIs, falls are prevalent. Demographics and risk factors have been documented in observational studies and cohorts, with proposed measures to reduce the frequency and morbidity of these accidents [28]. A significant correlation exists between free fall height and injury severity. Although it is not linear, greater fall heights correspond to higher Injury Severity Scores (ISSs) [29].

Our findings indicate that citywide efforts to prevent subway-related trauma should be explored further. Some strategies for preventing subway injuries include enhancing surveillance by station staff, job positions for staff to take care of elderly or disabled people, education programs, training, media intervention, limiting public access to subway tracks, and optimizing platform design. These measures can effectively reduce the incidence of injuries and promote media usage to prevent subway suicides as well as other types of suicides [15,30]. Unemployment is associated with a 2- to 3-fold increase in the relative risk of suicide. These studies could also assist in determining a specific unemployment rate that warrants heightened awareness and monitoring. Additionally, another effective strategy to reduce the occurrence of these tragic outcomes is to identify and target populations that are at higher risk [15]. Interventions targeting at-risk populations identified in this study may decrease the morbidity and mortality associated with subway-related trauma [11].

### 4.1. Strengths

Despite these limitations, our study possesses several significant strengths. One key strength is the inclusion of clinical outcome data, which sets it apart from many prior studies that focused only on injury patterns. Our data reveal important clinical outcomes such as discharge destinations, mortality rates, and the need for rehabilitation, offering a holistic view of patient recovery and post-trauma needs. The comprehensive inclusion of surgical interventions such as craniotomies and orthopedic procedures further underscores the severity of injuries and the importance of trauma care in improving patient outcomes.

The large sample size of 360 patients enhances the statistical power of the study, allowing for a more nuanced analysis of the different injury mechanisms (assaults, falls, suicide, train strikes, etc.) and their associated clinical outcomes. By exploring various injury types, the study provides valuable insights that can inform targeted prevention strategies and injury management protocols.

Additionally, the detailed demographic analysis, which covers age, gender, race, and ethnicity, is crucial for understanding how different populations are affected by subway-related trauma. This analysis is particularly important for tailoring public health interventions to vulnerable groups. Moreover, the study’s multidimensional approach, which examines a broad range of injury mechanisms, offers critical insights into how different accidents contribute to SRIs.

Finally, the ability to compare clinical outcomes, such as discharge destinations and mortality rates, with those of historical studies like the 1987 London Underground study adds a valuable dimension of context. This comparison allows for a deeper understanding of the evolving nature of trauma care and highlights the differences in clinical care and injury outcomes over time.

### 4.2. Limitations

Data were collected from the NTRACS Trauma Registry, which may not always provide complete or detailed information. This may limit the generalizability of the findings to other settings with different demographic and socioeconomic characteristics. A multicenter study incorporating all emergency medical services’ data would provide a more accurate description of injuries. These limitations suggest that further research, incorporating more comprehensive and geographically diverse data, is necessary to fully understand the epidemiology of SRIs in urban settings.

A limitation of this study is the potential misinterpretation from recoding categorical variables, especially for outcomes like discharge disposition and injury patterns. When mutually exclusive outcomes are recoded into binary indicators, unintended correlations may appear, as both categories can be flagged as “positive”, inflating associations.

While this study provides valuable insights into the injury patterns and clinical outcomes resulting from subway-related accidents, several limitations must be acknowledged. First, as a single-center retrospective study, these findings may not be generalizable to other trauma centers or urban settings. The patient population treated at one trauma center may not capture the full spectrum of SRIs across different regions or hospitals. This limits the broader applicability of the results.

Our study’s reliance on ICD injury descriptions and AIS body region classifications introduces the potential for bias. Differences in coding practices and documentation quality across clinicians can affect data accuracy, making it difficult to standardize injury descriptions or severity levels across all cases. Moreover, the retrospective nature of the study prevents the establishment of causal relationships between injury mechanisms and clinical outcomes, as it only captures information available in medical records.

Another limitation is the possibility of the underreporting of cases, particularly for patients with less severe injuries who may not seek immediate medical attention or those whose injuries were not documented due to inconsistent reporting practices. Therefore, the true incidence of SRIs is likely underestimated.

Finally, although our study provides valuable data on immediate clinical outcomes, it does not address the long-term consequences or complications of these injuries. Understanding the long-term impact on survivors’ quality of life, cognitive functioning, and healthcare utilization would offer a more comprehensive picture of subway-related trauma.

## 5. Conclusions

This study provides a comprehensive analysis of SRIs, highlighting a significant prevalence of traumatic head injuries among patients, with a notable incidence rate of 5.5%. The analysis reveals that falls are the leading cause of these injuries, accounting for 66.8% of cases, with older adults, particularly those over 65, comprising 24.5% of the patient population and demonstrating a concerning average age of 56 years.

While the overall mortality rate of 2.2% indicates that effective trauma management can minimize fatalities, the substantial proportion of patients who experience moderate to severe head injuries emphasizes the need for ongoing monitoring and rehabilitation efforts. The study also reports that 25% of patients required surgical intervention, reflecting the severity of these injuries and the critical need for timely and efficient medical care.

In summary, addressing the factors contributing to SRIs and focusing on preventive measures, particularly for vulnerable populations such as older adults, can prove to be essential in improving patient outcomes and enhancing overall safety within subway systems. Future research should focus on long-term effects and expand across multiple centers to capture a broader and more generalizable understanding of subway-related trauma.

## Figures and Tables

**Figure 1 life-15-00051-f001:**
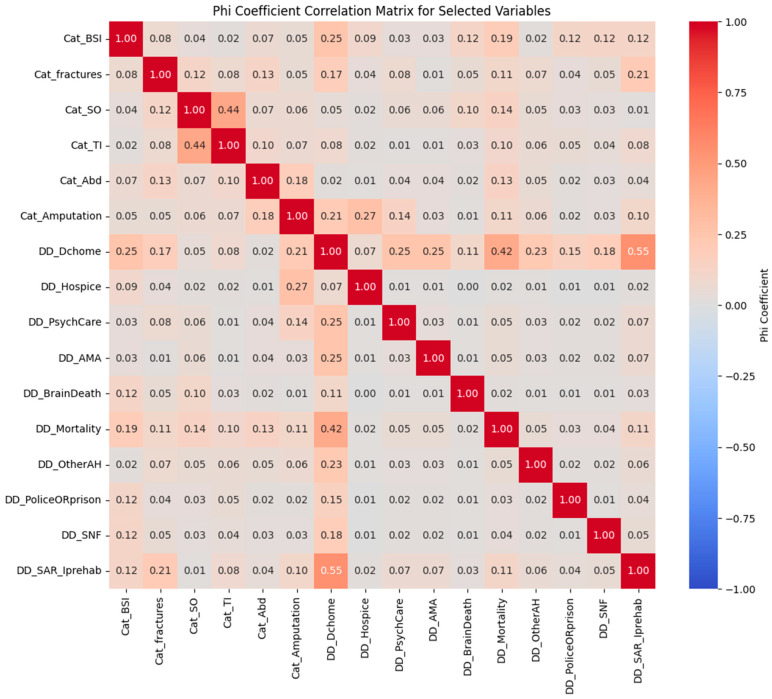
Correlation matrix for the Phi coefficient: injury patterns and hospital disposition in subway-related trauma.

**Table 1 life-15-00051-t001:** Distribution of trauma type and Injury Severity Score (ISS) groups among subway-related injuries (SRIs).

Descriptions		Count	Percentage
AIS per body region	AIS Head	89	24.7%
	AIS Thorax	60	16.7%
	AIS Low Ext	52	14.4%
	AIS Abdomen	14	3.9%
	AIS Spine	9	2.5%
Trauma Type	Blunt	338	93.9%
	Burn	1	0.3%
	Penetrating	21	5.8%
Injury Severity Score (ISS) group	Minor (ISS 1–9)	219	60.8%
	Moderate (ISS 10–16)	63	17.5%
	Severe (ISS17–24)	33	9.2%
	Very Severe (ISS ≥ 25)	45	12.5%
Overall ISS (Mean and SD)		10.69	11.33
Vent Days (Mean and SD)		0.77	3.10

Note: Percentages are based on the total number of patients with an AIS score ≥ 3 per body region, trauma type, and Injury Severity Score (ISS) group.

**Table 2 life-15-00051-t002:** Descriptives by mechanisms of injury (MOIs) in subway-related trauma.

Descriptives by Injury		Assault		Fall		Other (Electric, Struck Against Object, Other Injuries)		Suicide		Train-Struck		Overall	
		Count	%	Count	%	Count	%	Count	%	Count	%	Count	%
Gender	Female	4	1.1%	60	16.7%	1	0.3%	3	0.8%	3	0.8%	71	20%
	Male	50	13.9%	174	48.3%	4	1.1%	22	6.1%	39	10.8%	289	80%
Race	Asian	3	0.8%	33	9.2%	0	0.0%	1	0.3%	2	0.6%	39	11%
	Black	4	1.1%	36	10.0%	1	0.3%	1	0.3%	5	1.4%	47	13%
	Other	40	11.1%	118	32.8%	4	1.1%	13	3.6%	27	7.5%	202	56%
	Unknown	1	0.3%	8	2.2%	0	0.0%	5	1.4%	3	0.8%	17	5%
	White	6	1.7%	39	10.8%	0	0.0%	5	1.4%	5	1.4%	55	15%
Ethnicity	Hispanic Origin	35	9.7%	87	24.2%	3	0.8%	8	2.2%	20	5.6%	153	43%
	Non-Hispanic Origin	16	4.4%	130	36.1%	2	0.6%	11	3.1%	16	4.4%	175	49%
	Unknown	3	0.8%	17	4.7%	0	0.0%	6	1.7%	6	1.7%	32	9%
Age in years (Standard Deviation)		37.89 (15.95)		52.51 (17.81)		33.62 (18.63)		37.61 (13.54)		40.21 (13.31)		47.59 (18.05)	

**Table 3 life-15-00051-t003:** Injury patterns by MOI in subway-related trauma cases.

	Assault		Fall		Other (Electric, Struck Against Object, Other Injuries)		Suicide		Train-Struck	
	Count	%	Count	%	Count	%	Count	%	Count	%
Brain-specific injuries (subdural hemorrhage, epidural hemorrhage, subarachnoid hemorrhage, diffused traumatic brain injuries, and other intracranial injuries)	4	4.20%	63	65.60%	1	1.00%	11	11.50%	17	17.70%
Fractures (long bone extremities, facial bones, vertebral, metatarsals and metacarpal bones)	24	10.10%	160	67.50%	2	0.80%	18	7.60%	33	13.90%
Solid organ injuries (laceration/contusion of liver, spleen, lung, kidney, heart)	3	8.60%	23	65.70%	0	0.00%	3	8.60%	6	17.10%
Traumatic thoracic injuries (pneumothorax, hemothorax, pneumohemothorax, laceration, contusion and other unspecified injury of thorax)	11	28.90%	17	44.70%	1	2.60%	3	7.90%	6	15.80%
Abdominal injuries (laceration, contusion, puncture wound)	6	37.50%	3	18.80%	0	0.00%	1	6.30%	6	37.50%
Traumatic amputations (upper and lower extremity, other amputation)	1	7.70%	3	23.10%	1	7.70%	4	30.80%	4	30.80%

**Table 4 life-15-00051-t004:** Clinical outcomes of trauma patients by MOI.

	Assault		Fall		Other (Electric, Struck Against Object, Other Injuries)		Suicide		Train-Struck	
	Count	%	Count	%	Count	%	Count	%	Count	%
Discharge to a Home/Shelter	45	18.80%	174	72.50%	3	1.30%	3	1.30%	15	6.30%
Hospice	0	0.00%	0	0.00%	0	0.00%	0	0.00%	1	100.00%
Inpatient Psychiatry Care	1	9.10%	1	9.10%	0	0.00%	7	63.60%	2	18.20%
Left Against Medical Advice	1	9.10%	9	81.80%	0	0.00%	0	0.00%	1	9.10%
Met brain death criteria	0	0.00%	1	50.00%	0	0.00%	1	50.00%	0	0.00%
Mortality	2	6.90%	5	17.20%	0	0.00%	11	37.90%	11	37.90%
Other Acute Care Hospital Emergency Departments	1	11.10%	6	66.70%	1	11.10%	0	0.00%	1	11.10%
Police Custody/Jail/Prison	1	25.00%	2	50.00%	0	0.00%	0	0.00%	1	25.00%
Skilled Nursing Facility	0	0.00%	5	83.30%	0	0.00%	0	0.00%	1	16.70%
Sub-Acute Rehabilitation/Inpatient Rehabilitation	3	6.40%	31	66.00%	1	2.10%	3	6.40%	9	19.10%

**Table 5 life-15-00051-t005:** Emergency department (ED) disposition outcomes in trauma cases.

Emergency Department Disposition	Minor (ISS 1–9)	Moderate (ISS 10–16)	Severe (ISS 17–24)	Very Severe (ISS ≥ 25)	Overall
Admitted	152 (69.4%)	59 (93.7%)	32 (97.0%)	35 (77.8%)	278 (77.2%)
Died in ED	1 (0.5%)	1 (1.6%)	0 (0.0%)	4 (8.9%)	6 (1.7%)
Died on Arrival	1 (0.5%)	0 (0.0%)	1 (3.0%)	3 (6.7%)	5 (1.4%)
Died within 15 min	0 (0.0%)	1 (1.6%)	0 (0.0%)	3 (6.7%)	4 (1.1%)
Discharged to Home	59 (26.9%)	1 (1.6%)	0 (0.0%)	0 (0.0%)	60 (16.7%)
Direct Admission to Inpatient Ward	2 (0.9%)	0 (0.0%)	0 (0.0%)	0 (0.0%)	2 (0.6%)
Transferred to Another Hospital Emergency Dept.	4 (1.8%)	1 (1.6%)	0 (0.0%)	0 (0.0%)	5 (1.4%)

**Table 6 life-15-00051-t006:** This table presents analysis evaluated differences in Revised Trauma Scores (RTSs) across five categories of mechanisms of injury (MOIs): Assault, Fall, Other, Suicide, and Train-struck.

Statistic	Value					
F-statistic	28.70					
*p*-value	1.119e-20					
MOI Group 1	MOI Group 2	Mean Difference	*p*-adj (Adjusted *p*-value)	Lower Bound	Upper Bound	Significant
Assault	Fall	−0.0345	0.9998	−0.5908	0.5219	No
Assault	Other	−2.2759	0.0003	−3.7466	−0.8051	Yes
Assault	Suicide	−2.0146	0	−2.9019	−1.1273	Yes
Assault	Train-struck	−1.7498	0	−2.5104	−0.9891	Yes
Fall	Other	−2.2414	0.0002	−3.6442	−0.8386	Yes
Fall	Suicide	−1.9801	0	−2.7496	−1.2106	Yes
Fall	Train-struck	−1.7153	0	−2.3345	−1.0961	Yes
Other	Suicide	0.2613	0.9909	−1.3026	1.8252	No
Other	Train-struck	0.5261	0.8709	−0.9696	2.0217	No
Suicide	Train-struck	0.2648	0.9356	−0.6632	1.1928	No

**Table 7 life-15-00051-t007:** This table summarizes the relationships between the mechanisms of injury (MOIs) and various medical procedures.

Procedure	Chi-Square	*p*-Value	Degrees of Freedom	Significant
Neurosurgery Intervention	11.68013646	0.019895325	4	Yes
Orthopedic Procedure	5.218210789	0.265631448	4	No
Intubation	43.76031065	7.19E-09	4	Yes
Transfusion	0.889460287	0.92606802	4	No
Mechanical Ventilation Days	35.88748451	0.956707801	52	No
Vascular Access Insertion	1.134764219	0.888717979	4	No
Drainage Intervention	2.96828205	0.563147279	4	No

## Data Availability

The data were requested from the Elmhurst trauma registry and extracted using electronic medical records after receiving approval from the Institutional Review Board at our facility (Elmhurst Hospital Center).

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
