# Peer review of "Clinical Outcomes and Patterns of Traumatic Injuries Associated with Subway Incidents at a Level 1 Trauma Center"

_life, 2025, doi:10.3390/life15010051_

Round 1
Reviewer 1 Report
Comments and Suggestions for Authors
Dear Authors,
I read with great interest your manuscript about the trauma burden in cases of metro incidents.
However, some aspects require your attention.
Suicide cases become more complex with each year and there is an increasing trend of self-inflicted trauma at the level of the neck. Please reference this to the article by Vrinceanu D, Banica B, Papacocea R, Papacocea T. Self-inflicted laryngeal penetrating wounds with suicidal intention: two clinical cases. Rom J Leg Med. 2018; 26:16–20.
There are many abbreviations in the text, please insert a list of abbreviations at the end of the manuscript.
Please update the references. There are titles from 1987 and 1988. I am sure there are newer titles on the subject that are worth mentioning.
Moreover, format the references according to MDPI instructions for authors.
Looking forward to receiving the improved version of your manuscript.
Author Response
Dear reviewers,
Thank you for providing these comments. We have carefully considered each comment to respond as effectively as possible. Please see our point-by-point response below
Reviewer 1
- Suicide cases become more complex with each year and there is an increasing trend of self-inflicted trauma at the level of the neck. Please reference this to the article by Vrinceanu D, Banica B, Papacocea R, Papacocea T. Self-inflicted laryngeal penetrating wounds with suicidal intention: two clinical cases. Rom J Leg Med. 2018; 26:16–20.
Response: We have included this reference in the manuscript
- There are many abbreviations in the text, please insert a list of abbreviations at the end of the manuscript.
Response: All abbreviations have been inserted at the end of the manuscript
Abbreviations:
ICD- International Classification of Diseases
AIS-Abbreviated Injury Scale
TBI- Traumatic Brain Injury
SAH- Subarachnoid Hemorrhage
EDH- Epidural Hemorrhage
SDH- Subdural Hemorrhage
ISS- Injury Severity Score
SRIs- Subway Related Injuries
LOS- Length of Stay
ICU- Intensive Care Unit
NTRACS-National Trauma Registry of the American College of Surgeons
MOI- Mechanism of Injury
ED- Emergency Department
PHI- Protected Health Information
- Please update the references. There are titles from 1987 and 1988. I am sure there are newer titles on the subject that are worth mentioning.
Response: There were 2 references 1987 and 1988 and we have replaced them with other latest references.
- Moreover, format the references according to MDPI instructions for authors.
Response: References have been formatted according to MDPI instructions for authors
We hope our responses meet your expectations and we look forward to your feedback.
Sincerely,
Bharti Sharma (Corresponding author)
Research Associate,
Research Coordinator,
Department of Surgery,
Mount Sinai Health System
Elmhurst Hospital Center
NYC Health + Hospitals/Elmhurst,
79-01 Broadway, New York, Queens, NY 11373

Reviewer 2 Report
Comments and Suggestions for Authors
Overview:
The authors performed a retrospective, descriptive study of subway related injuries seen at a single trauma center, covering patients from January 1, 2016, to December 31, 2023. Patients were identified using International Classification of Diseases (ICD) injury descriptions and Abbreviated Injury Scale (AIS) body region information. Out of 360 patients (total), 23.5% presented with head injuries with an AIS score ≥ 3. Patients with blunt trauma (93.99 %) were more common than penetrating (5.74%) and burn trauma (0.3%). Overall, the mean Injury Severity Score (ISS) was 10.69. ISS for severe injuries (17-24) comprised 9.2%, moderate injuries (10-16) 17.5%, and minor injuries (1-9) 60.8% of cases. Falls had the highest percentage of traumatic brain injury (TBI) (65.60%) and fractures (67.50%). Assaults were associated with thoracic injuries (28.90%). Suicide attempts demonstrated a high percentage of traumatic amputations (30.80%). In the emergency department (ED), most patients (69.4%) were admitted for further care, such as trauma, neurosurgery, or others. 0.5% died in the ED, 0.5% died on arrival, and 1.04% died within 15 minutes of arrival.
Introduction:
The authors describe prior descriptive studies and propose to update the literature on subway injury descriptive injuries. They suggest that injury patterns can aid the workup and management of these injuries. It would be helpful if the investigators also looked for potential preventive measures for more common subway injuries.
Methods:
Given that data were retrieved from the National Trauma Registry of the American College of Surgeons (NTRACS), it is unclear why the authors focused only on cases seen at one trauma center. Did they use other data sources from that trauma center to enrich their analysis?
Data were tabulated by injury type, severity, disposition, demographics, and mechanism of injury.
Results:
Multiple tables are presented.
Lines 197-201: A correlation matrix of injury patterns and hospital dispositions is provided. “…the correlation matrix highlights expected clinical patterns, such as the relationship between discharge destinations and patient stability, while also revealing the importance of how data is coded, especially for mutually exclusive outcomes like discharge to home and rehab. This correlation analysis allows clinicians to better anticipate patient needs post-discharge and make informed decisions about resource allocation.”
It would be interesting to know if unusual correlations led to reconsideration of care plans or discharge considerations.
Discussion:
The addition of comments regarding potential injury mitigation to minimize falls (the most common injury mechanism) would be a nice addition to this report.
Lines 311-339:
It is unclear why there are limitations described here plus in a separate “4.2 Limitations” section on lines 366-380. It would be better to move the information from lines 311-339 to the limitations section.
Lines 319-324:
While the authors are correct that a single institution analysis can be a limitation, they should also point out that the similarity of their data with the Bellevue study suggests some generalizability to NYC subway injuries managed elsewhere in NYC.
Conclusions:
Lines 391-395:
The authors state that “In summary, addressing the factors contributing to SRIs and focusing on preventive measures, particularly for vulnerable populations such as older adults, can prove to be essential in improving patient outcomes and enhancing overall safety within subway systems.”
As noted previously, it would strengthen this manuscript if the authors speculated on potential injury mitigation measures based upon this observational study. For example, how might older adults be protected from falls in SRIs?
Author Response
Dear reviewer,
Thank you for providing these comments. We have carefully considered each comment to respond as effectively as possible. Please see our point-by-point response below
Reviewer 2
Response: References have been changed based on MDPI instructions for authors
Introduction:
- The authors describe prior descriptive studies and propose to update the literature on subway injury descriptive injuries. They suggest that injury patterns can aid the workup and management of these injuries. It would be helpful if the investigators also looked for potential preventive measures for more common subway injuries.
Methods:
- Given that data were retrieved from the National Trauma Registry of the American College of Surgeons (NTRACS), it is unclear why the authors focused only on cases seen at one trauma center. Did they use other data sources from that trauma center to enrich their analysis?
Response: Our Hospital (Elmhurst Hospital Center) trauma registry utilizes NTRACS software (American College of Surgeons National Trauma Registry System). We requested data from the Trauma registry at EHC including PHI and PHIs were used for patient chart review (wherever applicable). So, we used 2 sources of data: data from the trauma registry and the Patient’s medical chart.
Multiple tables are presented.
Lines 197-201: A correlation matrix of injury patterns and hospital dispositions is provided. “…the correlation matrix highlights expected clinical patterns, such as the relationship between discharge destinations and patient stability, while also revealing the importance of how data is coded, especially for mutually exclusive outcomes like discharge to home and rehab. This correlation analysis allows clinicians to better anticipate patient needs post-discharge and make informed decisions about resource allocation.”
- It would be interesting to know if unusual correlations led to a reconsideration of care plans or discharge considerations.
Response: (Lines 174-177) The Phi coefficient correlation matrix provides valuable insights into associations between various binary clinical outcomes, informing patient care planning and resource allocation. Although discharge to home and discharge to subacute rehab are theoretically mutually exclusive options. Trauma patients were usually discharged to either one or the other, not both.
(Lines 187-196) There was a moderate positive correlation between amputation and discharge to hospice care (0.27), indicating that patients who undergo amputation may often be dealing with severe or end-of-life conditions and poor prognosis, underscoring the importance of hospice care planning for these patients. Negative associations were also notable, particularly between discharge to home and both mortality (-0.42). These associations imply that patients discharged to home are generally in more stable conditions. Many other variable pairs showed weak or minimal correlations, such as solid organ injury patterns and psychiatric care discharge disposition outcomes, indicating that these factors may operate independently in this dataset.
(Lines 281-283) Our results also show that timely and effective trauma care likely contributed to the favorable outcomes observed in many patients, with a large proportion of fall patients (72.5%) being discharged home or to a shelter.
Discussion:
- The addition of comments regarding potential injury mitigation to minimize falls (the most common injury mechanism) would be a nice addition to this report.
Response: (lines 351-362) Our findings indicate that citywide efforts to prevent subway-related trauma should be explored further. Some strategies for preventing subway injuries include enhancing surveillance by station staff, job positions for staff to take care of elderly or disabled people, education programs, training, media intervention, limiting public access to subway tracks, and optimizing platform design. These measures can effectively reduce the incidence of injuries and help manage media coverage of subway suicides (14). Unemployment is associated with a 2- to 3-fold increase in the relative risk of suicide. These studies could also assist in determining a specific unemployment rate that warrants heightened awareness and monitoring. Additionally, another effective strategy to reduce the occurrence of these tragic outcomes is to identify and target populations that are at higher risk (14). Interventions targeting at-risk populations identified in this study may decrease the morbidity and mortality associated with subway-related trauma (11).
- Lines 311-339:
It is unclear why there are limitations described here plus in a separate “4.2 Limitations” section on lines 366-380. It would be better to move the information from lines 311-339 to the limitations section.
Response: We moved the information from lines 311-339 to the limitations section.
Data was collected from the NTRACS Trauma Registry, which may not always provide complete or detailed information. This may limit the generalizability of the findings to other settings with different demographic and socioeconomic characteristics. A multicenter study incorporating all emergency medical services’ data would provide a more accurate description of the injuries. These limitations suggest that further research, incorporating more comprehensive and geographically diverse data, is necessary to fully understand the epidemiology of SRIs in urban settings.
A limitation of this study is the potential misinterpretation from recoding categorical variables, especially for outcomes like discharge disposition and injury patterns. When mutually exclusive outcomes are recoded into binary indicators, unintended correlations may appear, as both categories can be flagged as “positive,” inflating associations.
While this study provides valuable insights into the injury patterns and clinical outcomes resulting from subway-related accidents, several limitations must be acknowledged. First, as a single-center retrospective study, the findings may not be generalizable to other trauma centers or urban settings. The patient population treated at one trauma center may not capture the full spectrum of SRIs across different regions or hospitals. This limits the broader applicability of the results.
Our study's reliance on ICD injury descriptions and AIS body region classifications introduces the potential for bias. Differences in coding practices and documentation quality across clinicians can affect data accuracy, making it difficult to standardize injury descriptions or severity levels across all cases. Moreover, the retrospective nature of the study prevents the establishment of causal relationships between injury mechanisms and clinical outcomes, as it only captures information available in medical records.
Another limitation is the possibility of underreporting of cases, particularly for patients with less severe injuries who may not seek immediate medical attention or those whose injuries were not documented due to inconsistent reporting practices. Therefore, the true incidence of SRIs is likely underestimated.
Finally, although our study provides valuable data on immediate clinical outcomes, it does not address the long-term consequences or complications of these injuries. Understanding the long-term impact on survivors’ quality of life, cognitive functioning, and healthcare utilization would offer a more comprehensive picture of subway-related trauma.
- Lines 319-324:
While the authors are correct that a single institution analysis can be a limitation, they should also point out that the similarity of their data with the Bellevue study suggests some generalizability to NYC subway injuries managed elsewhere in NYC.
Response: (line # 240-242) Furthermore, our study exhibits that fractures were predominantly associated with falls (67.50%), aligning with the Bellevue Hospital study data that indicates a 49% incidence of at least one fracture among patients.
Conclusions:
Lines 391-395:
The authors state that “In summary, addressing the factors contributing to SRIs and focusing on preventive measures, particularly for vulnerable populations such as older adults, can prove to be essential in improving patient outcomes and enhancing overall safety within subway systems.”
- As noted previously, it would strengthen this manuscript if the authors speculated on potential injury mitigation measures based on this observational study. For example, how might older adults be protected from falls in SRIs?
Response: (lines 351-362) Our findings indicate that citywide efforts to prevent subway-related trauma should be explored further. Some strategies for preventing subway injuries include enhancing surveillance by station staff, job positions for staff to take care of elderly or disabled people, education programs, training, media intervention, limiting public access to subway tracks, and optimizing platform design. These measures can effectively reduce the incidence of injuries and help manage media coverage of subway suicides (14). Unemployment is associated with a 2- to 3-fold increase in the relative risk of suicide. These studies could also assist in determining a specific unemployment rate that warrants heightened awareness and monitoring. Additionally, another effective strategy to reduce the occurrence of these tragic outcomes is to identify and target populations that are at higher risk (14). Interventions targeting at-risk populations identified in this study may decrease the morbidity and mortality associated with subway-related trauma (11).
We hope our responses meet your expectations and we look forward to your feedback.
Sincerely,
Bharti Sharma (Corresponding author)
Research Associate,
Research Coordinator,
Department of Surgery,
Mount Sinai Health System
Elmhurst Hospital Center
NYC Health + Hospitals/Elmhurst,
79-01 Broadway, New York, Queens, NY 11373

Reviewer 3 Report
Comments and Suggestions for Authors
The authors present a manuscript describing clinical outcomes and patterns associated with subway incidents at a Level 1 Trauma Center. The manuscript is well conceived and written, clearly describing the injury types and outcomes. Overall, it is very well done.
One suggestion-It would be helpful if you can propose any prevention ideas based on the type of injuries observed.
Author Response
Dear reviewer,
Thank you for providing these comments. We have carefully considered each comment to respond as effectively as possible. Please see our point-by-point response below
Reviewer 3
The authors present a manuscript describing clinical outcomes and patterns associated with subway incidents at a Level 1 Trauma Center. The manuscript is well conceived and written, clearly describing the injury types and outcomes. Overall, it is very well done.
- One suggestion- It would be helpful if you could propose any prevention ideas based on the type of injuries observed.
Response: (lines 351-362) Our findings indicate that citywide efforts to prevent subway-related trauma should be explored further. Some strategies for preventing subway injuries include enhancing surveillance by station staff, job positions for staff to take care of elderly or disabled people, education programs, training, media intervention, limiting public access to subway tracks, and optimizing platform design. These measures can effectively reduce the incidence of injuries and help manage media coverage of subway suicides (14). Unemployment is associated with a 2- to 3-fold increase in the relative risk of suicide. These studies could also assist in determining a specific unemployment rate that warrants heightened awareness and monitoring. Additionally, another effective strategy to reduce the occurrence of these tragic outcomes is to identify and target populations that are at higher risk (14). Interventions targeting at-risk populations identified in this study may decrease the morbidity and mortality associated with subway-related trauma (11).
We hope our responses meet your expectations and we look forward to your feedback.
Sincerely,
Bharti Sharma (Corresponding author)
Research Associate,
Research Coordinator,
Department of Surgery,
Mount Sinai Health System
Elmhurst Hospital Center
NYC Health + Hospitals/Elmhurst,
79-01 Broadway, New York, Queens, NY 11373

Reviewer 4 Report
Comments and Suggestions for Authors
1.What is the study hypothesis? What is the PICO questions?
2. The primary and subsequent study objectives remains unclear. Please be more specific about the aim of the research.
3.Please present more detailed the study methodology-data collection, analysis protochol
4.You used AIS (that is an anatomic score, and ISS. It is not very clear how did You processed patients medical data only with this scores, without TS (trauma score), RTS (revised trauma score) or others dinamic indicator of vital function in different stages on medical intervention?
5. Please argument how relevant is the ICD to assess, and compare the medical outcome of different diseases
6. Please include informations about the trauma team competencies (medical teams, paramedical/EMT), and homogenity acting in the prehospital stage of interventions - as a predicators of medical outcome
7. Please explain data from Table no 5 - patients with minor or moderate injuries died in ED, on arrival, or within 15 minutes?
8. Please include în analysis the triage codes of the patients related to presentation accuity în ED
9. Please reconsider final conclusions including only main and relevant results, further subsequent research or benefits for medical practice.
10. Please reconsider the references - include new references (preferably from 5 years)
Author Response
Dear reviewer,
Thank you for providing these comments. We have carefully considered each comment to respond as effectively as possible. Please see our point-by-point response below
Reviewer 4
- What is the study hypothesis? What is the PICO questions (patient/population, intervention, comparison and outcomes)?
Response: (lines 65-71) We hypothesize that subway injuries may lead to various traumatic injury patterns along with poor clinical outcomes and our objective is to study various traumatic injury patterns associated with subway traumatic incidents and related clinical outcomes. By delving into the specific injury patterns seen in subway-related incidents, particularly within the context of a level 1 trauma center, this study seeks to expand the current understanding of these complex cases and that of the development of more effective clinical outcomes.
PICO: This is a retrospective study from a single center, including patients with subway-related injuries (SRIs) from January 1, 2016, to December 31, 2023. It is not a comparative or an intervention study. We have studied various clinical outcomes associated with SRIs.
- The primary and subsequent study objectives remain unclear. Please be more specific about the aim of the research.
Response: (Lines 69-76) There is limited published data from the United States regarding subway injuries. Hence, Our primary objective is to study various traumatic injury patterns and mechanisms of injury associated with subway traumatic incidents whereas our secondary objective is to study various clinical outcomes/discharge dispositions associated with these injury patterns, example Discharge Home/Shelter, Hospice, Inpatient Psychiatry Care, Left Against Medical Advice, Met brain death criteria, Mortality, Other Acute Care Hospital Emergency Department (ED) or, Police Custody/Jail/Prison, Skilled Nursing Facility, and Sub Acute Rehabilitation / In-patient Rehabilitation.
- Please present more detailed about the study methodology-data collection, analysis protocol
Response: We have revised the method section.
This is a single-center, retrospective review conducted at a level 1 trauma center verified by the American College of Surgeons in Queens, New York City. We included all patients who presented with traumatic SRIs between January 1, 2016- December 31, 2023, inclusive. All patients with subway-related injuries were included. We excluded patients with no subway-related injuries.
Patient data were requested from the Trauma Registry at our facility (Elmhurst Hospital Center). Our center’s trauma registry utilizes NTRACS software (American College of Surgeons National Trauma Registry System). Patients were identified based on the injury mechanism, cause of injury, primary mechanisms (lCD9 or lCDL0 E-Code), and the Abbreviated Injury Severity (AIS) score. The AIS score ranges from 1 to 6 per body region.
Utilizing trauma registry data and adhering to inclusion and exclusion criteria, we identified a final patient group of 360 individuals. The medical charts of the patients were reviewed, and all relevant information required for this study was collected.
We collected data using a data collection tool (Excel sheet or spreadsheet). We incorporated all data elements into this tool. Examples of data elements are demographics (for example: age, sex, race, ethnicity), AIS, GCS, ISS, injury pattern, discharge disposition, mortality status, mechanism of injury (MOI), and others. The dataset underwent several preprocessing steps to ensure data integrity, confidentiality, and suitability for statistical analysis. Unique identifiers, including medical record numbers (MRNs), dates of birth (DOBs), and patient names, were removed to deidentify the dataset. This process was carried out to maintain patient privacy in compliance with ethical research standards.
The Abbreviated Injury Severity (AIS) regions for patients were evaluated, with regions of interest being head, face, neck, thorax, abdomen, spine, upper extremity, lower extremity, and external injuries. Furthermore, the mechanisms of injury were studied: As-sault, Fall, Suicide, and Train struck. The types of subway-related trauma were analyzed under the mechanisms of injury. The types of subway-related trauma include the sum of facial fractures, vertebral fractures, SAH, SDH, Skull, Solid Organ Injury, Traumatic thoracic injuries, EDH, Hand/Foot fractures, Abdominal Injury, and diffuse TBI. Also, gender, race, age, and ethnicity data were collected under the mechanisms of injury.
Furthermore, the clinical outcomes were collected under the mechanisms of injury. The clinical outcomes include Discharge Home/Shelter, Hospice, Inpatient Psychiatry Care, Left Against Medical Advice, Met brain death criteria, Mortality, Other Acute Care Hospital Emergency Department (ED) or, Police Custody/Jail/Prison, Skilled Nursing Facility, and Sub Acute Rehabilitation / Inpatient Rehabilitation.
For Statistical analysis, we used the Phi coefficient correlation matrix to examine relationships between trauma injury patterns associated with subway trauma and clinical outcomes based on discharge disposition. Injury trends and discharge outcomes, such as discharge to a skilled nursing facility, home, subacute rehabilitation, and hospice, were abstracted for the electronic medical record.
First, we evaluated connections using the Chi-Square test. However, this approach was limited by assumptions that were not met, such as the mutual exclusion of some dis-charge dispositions and low predicted frequencies in some cells. We recorded all variables as binary indicators to better capture associations and computed Phi coefficients for every pair. A heatmap displayed the results, highlighting important connections between injury trends and discharge outcomes that can guide post-discharge care and resource allocation. Further description about these statistical approaches are elaborated within result section to avoid duplication.
4.You used AIS (that is an anatomic score, and ISS. It is not very clear how did You processed patients medical data only with this scores, without TS (trauma score), RTS (revised trauma score) or others dinamic indicator of vital function in different stages on medical intervention?
Response- Patient data was requested from the National Trauma Registry of the American College of Surgeons (NTRACS) Database at our center (Elmhurst Hospital Center). We identified Patients based on the injury mechanism, cause of injury, primary mechanisms (lCD9 or lCDL0 E-Code), and the Abbreviated Injury Severity (AIS) score.
- Please argument how relevant is the ICD to assess, and compare the medical outcome of different diseases
Response- Trauma registry at our center maintains primary mechanisms (lCD9 or lCDL0 E-Code) and these codes were used as one of the key identifier in this study. We are not sure about comparing medical outcomes of other diseases because that is not the main focus of our study.
- Please include informations about the trauma team competencies (medical teams, paramedical/EMT), and homogenity acting in the prehospital stage of interventions - as a predicators of medical outcome
Response-(line # 319-323) Paramedics play an essential role in patient care. Quality of care provided by medical teams, paramedical/EMT can be affected by patient’s critical condition or severity of injury. For example, prehospital intubation results in higher mortality than in patient who are non-prehospital intubated. Intervention such as intubation might result into negative outcomes.
- Please explain data from Table no 5 - patients with minor or moderate injuries died in ED, on arrival, or within 15 minutes?
Response: The majority of patients with SRIs (69.4%) were admitted for further care, while a smaller percentage experienced fatal outcomes, with 0.5% dying in the ED, 0.5% dying on arrival, and 1.04% dying within 15 minutes of arrival to the ED.
- Please include în analysis the triage codes of the patients related to presentation accuity în ED
Response: Our registry do not maintain records based on triage colors. But we have include patients but we have included all possible information on injury patterns, mechanism of injury, discharge disposition, mortality. Our tables and results (please refer to table 5) include analysis on all triage codes. Category I-viable victims for example: who Died within 15 minutes, Died on Arrival (table 5). Category II: Used for victims with non-life-threatening injuries, but who urgently require treatment for example Admitted (directly or transferred) (table 5). Category III: Used for victims with minor injuries for example Discharged to Home (table 5).
- Please reconsider final conclusions including only main and relevant results, further subsequent research or benefits for medical practice.
Response: We have included all major points as well as need for future research.
This study provides a comprehensive analysis of SRIs highlighting a significant prevalence of traumatic head injuries among patients, with a notable incidence rate of 5.5%. The analysis reveals that falls are the leading cause of these injuries, accounting for 66.8% of cases, with older adults, particularly those over 65, comprising 24.5% of the patient population and demonstrating a concerning average age of 56 years.
While the overall mortality rate of 2.2% indicates that effective trauma management can minimize fatalities, the substantial proportion of patients who experience moderate to severe head injuries emphasizes the need for ongoing monitoring and rehabilitation efforts. The study also reports that 25% of patients required surgical intervention, reflecting the severity of these injuries and the critical need for timely and efficient medical care.
In summary, addressing the factors contributing to SRIs and focusing on preventive measures, particularly for vulnerable populations such as older adults, can prove to be essential in improving patient outcomes and enhancing overall safety within subway systems. Future research should future research should focus on long-term effects and expand across multiple centers to capture a broader and more generalizable understanding of subway-related trauma.
- Please reconsider the references - include new references (preferably from 5 years)
Response: We have modified our references. Many are from 2024, 2023, 2021, 2018. Also, there were 2 references 1987 and 1988 and we have replaced them with other latest references.
We hope our responses meet your expectations and we look forward to your feedback.
Sincerely,
Bharti Sharma (Corresponding author)
Research Associate,
Research Coordinator,
Department of Surgery,
Mount Sinai Health System
Elmhurst Hospital Center
NYC Health + Hospitals/Elmhurst,
79-01 Broadway, New York, Queens, NY 11373

Round 2
Reviewer 4 Report
Comments and Suggestions for Authors
Dear Authors,
Some of the earlier inconsistencies still persist:
1. Please do an in-depht comparative analysis of patients groups regarding the trauma scores, not only anatomic. The prognosis is strong related to the evolution of vital functions, so that, initial assessment data, secondary and tertiary survey relations is crucial.
2. Please explain the results from Tabel no 2, about the mortality în group with low ISS (1-9)
3. Please explain the Table no 3 în which lung and hearth injuries are classified differently from others chest trauma. At the same time the framing of the fracture type injuries include long bone injuries as well as facial bone fractures. It is well know that the hemodynamic and airway impact of different bone fractures are significantly different.
4. Please be more detailed about the medical interventions, the moment of intubation, pelvic splint, vascular access ( for example), damage control methods, time to vital functions stabilization, time to surgical intervention. All these have impact on survival and rehabilitation chance, and relationship with outcome.
Author Response
Dear reviewer,
Thank you so much for providing these comments. We have tried to respond to all of them. Please find point to point responses below:
- Please do an in-depth comparative analysis of patient’s groups regarding the trauma scores, not only anatomic. The prognosis is strong related to the evolution of vital functions, so that, initial assessment data, secondary and tertiary survey relations is crucial.
Response: As per your suggestions we performed in depth comparative analysis and introduced a new table as Table # 6. The Revised Trauma Score (RTS), which incorporates GCS, SBP, and RR, was calculated for each patient. RTS was coded into the dataset as a composite score representing the physiological severity of trauma.
|
Statistics |
Value |
|||||
|
F-statistic |
28.70 |
|||||
|
P-value |
1.119e-20 |
|
|
|
|
|
|
MOI Group 1 |
MOI Group 2 |
Mean Difference |
P-adj (Adjusted p-value) |
Lower Bound |
Upper Bound |
Significant |
|
Assault |
Fall |
-0.0345 |
0.9998 |
-0.5908 |
0.5219 |
No |
|
Assault |
Other |
-2.2759 |
0.0003 |
-3.7466 |
-0.8051 |
Yes |
|
Assault |
Suicide |
-2.0146 |
0 |
-2.9019 |
-1.1273 |
Yes |
|
Assault |
Trainstruck |
-1.7498 |
0 |
-2.5104 |
-0.9891 |
Yes |
|
Fall |
Other |
-2.2414 |
0.0002 |
-3.6442 |
-0.8386 |
Yes |
|
Fall |
Suicide |
-1.9801 |
0 |
-2.7496 |
-1.2106 |
Yes |
|
Fall |
Trainstruck |
-1.7153 |
0 |
-2.3345 |
-1.0961 |
Yes |
|
Other |
Suicide |
0.2613 |
0.9909 |
-1.3026 |
1.8252 |
No |
|
Other |
Trainstruck |
0.5261 |
0.8709 |
-0.9696 |
2.0217 |
No |
|
Suicide |
Trainstruck |
0.2648 |
0.9356 |
-0.6632 |
1.1928 |
No |
The analysis evaluated differences in Revised Trauma Score (RTS) across five categories of Mechanism of Injury (MOI): Assault, Fall, Other, Suicide, and Trainstruck. RTS is a validated tool that assesses physiological derangements in trauma patients and predicts survival outcomes. It incorporates key indicators such as vital signs and Glasgow Coma Scale scores, making it a critical metric for trauma assessment and management.
A one-way analysis of variance (ANOVA) revealed a statistically significant effect of MOI on RTS (F(4,N)=28.70,p<0.001F(4, N) = 28.70, p < 0.001F(4,N)=28.70,p<0.001), indicating that trauma severity varies significantly across the different mechanisms of injury. To further explore these differences, pairwise comparisons were conducted using Tukey’s Honest Significant Difference (HSD) test. Tukey’s HSD applies adjusted p-values (P-adj) to control the family-wise error rate, mitigating the risk of Type I errors that increase with multiple comparisons. This ensures the reliability and robustness of statistical inferences.
Post hoc analysis identified significant differences between groups. RTS scores for Assault were significantly lower than those for Other, Suicide, and Trainstruck (P−adj<0.001P-adj < 0.001P−adj<0.001), indicating greater trauma severity associated with assaults. Similarly, RTS scores for Fall were significantly lower compared to Other, Suicide, and Trainstruck (P−adj<0.001P-adj < 0.001P−adj<0.001). However, no significant differences were observed between Other and Suicide (P−adj=0.991P-adj = 0.991P−adj=0.991), Other and Trainstruck (P−adj=0.871P-adj = 0.871P−adj=0.871), or Suicide and Trainstruck (P−adj=0.936P-adj = 0.936P−adj=0.936). These findings suggest that while some mechanisms, such as Assault and Fall, are associated with higher trauma severity, others (e.g., Other, Suicide, and Trainstruck) may result in comparable physiological impacts as measured by RTS.
The adjusted p-values in Tukey’s HSD analysis were critical to ensuring the validity of pairwise comparisons. By accounting for the increased likelihood of false positives due to multiple comparisons, the adjusted p-values provide a rigorous framework for identifying true differences between groups. This approach reinforces the statistical integrity of the findings and ensures that significant results are not merely artifacts of chance.
Clinically, the results highlight the heightened trauma severity associated with mechanisms like Assault and Fall, underscoring the need for targeted interventions during triage and resource allocation. Conversely, the comparable RTS scores among Other, Suicide, and Trainstruck suggest that these mechanisms may require similar clinical approaches. These findings affirm the utility of RTS as a critical tool for trauma care and provide valuable insights for optimizing outcomes based on the mechanism of injury. The use of adjusted p-values further enhances the reliability of these conclusions, making them robust for both clinical and research applications.
Line # 372-377: Findings based on table 6 has been explained in result and discussion section of our manuscript.
We have also included it in our method section.
Line # 135-142: The Revised Trauma Score (RTS), which incorporates Glasgow coma scale (GCS), systolic blood pressure (SBP), and respiratory rate (RR), was calculated for each patient. For calculation we used formula: RTS = 0.9368 GCS + 0.7326 SBP + 0.2908 RR. RTS was coded into the dataset as a composite score representing the physiological severity of trauma. By integrating the revised trauma score (RTS) calculation, mean imputation for missing data, and Chi-square tests for categorical variables, this study comprehensively examines subway-related trauma patterns and their associated clinical outcomes.
- Please explain the results from Table no 2, about the mortality în group with low ISS (1-9)
Response: Yes, we have incorporated the explanation on mortality in group with low ISS (1-9) within manuscript.
Line # 152-158: The distribution of the ISS provides insights into the frequency of injuries based on their severity levels within the study population. Overall, the mean ISS of 10.69, with a standard deviation of 11.33, suggests a broad range of traumatic injuries. Most patients (suffered from minor injuries, the ISS of which is between 1 and 9. In total, these patients constitute 60.8% (219 cases). Meanwhile, around 17.5% (63 cases) were observed with moderate injuries, the ISS score between 10 and 16. Severe injuries (the ISS of 17-24) comprised 9.2% of the study cohort (33 cases).
- Please explain the Table no 3 in which lung and hearth injuries are classified differently from others chest trauma. At the same time the framing of the fracture type injuries includes long bone injuries as well as facial bone fractures. It is well known that the hemodynamic and airway impact of different bone fractures are significantly different.
Response: We appreciate your feedback. Based on the records of Trauma registry at our center Heart, lung, kidney, liver, spleen is included under solid organs. Hence, we chose to keep all 5 of them under solid organ injury.
As mentioned throughout, our manuscript is focused on various types of injury pattern. In our patient population, we found various types of fractures and analyzed it together. As Fracture is a single type of injury pattern so in order to make entire data sensible and organized, we analyzed and reported all types of fractures together.
- Please be more detailed about the medical interventions, the moment of intubation, pelvic splint, vascular access (for example), damage control methods, time to vital functions stabilization, time to surgical intervention. All these have impact on survival and rehabilitation chance, and relationship with outcome.
Response: Thanks for your comment. Yes, we have introduced a section focusing on medical interventions and a new table as Table 7
|
Procedure |
Chi-square |
p-value |
Degrees of Freedom |
Significant |
|
Neurosurgery Intervention |
11.68013646 |
0.019895325 |
4 |
Yes |
|
Orthopedic Procedure |
5.218210789 |
0.265631448 |
4 |
No |
|
Intubation |
43.76031065 |
7.19E-09 |
4 |
Yes |
|
Transfusion |
0.889460287 |
0.92606802 |
4 |
No |
|
Mechanical Ventilation Days |
35.88748451 |
0.956707801 |
52 |
No |
|
Vascular Access Insertion |
1.134764219 |
0.888717979 |
4 |
No |
|
Drainage Intervention |
2.96828205 |
0.563147279 |
4 |
No |
The relationships between the mechanism of injury (MOI) and various procedures were analyzed using the Chi-square test, a suitable method for evaluating associations between categorical variables like the mechanism of injury and binary medical procedure outcomes. The test identifies whether the observed distribution deviates significantly from chance.
Significant associations were found for NSGY Intervention p=0.0199 and Intubation p<0.00001, indicating their likelihood is related to the mechanism of injury. No significant associations were found for Orthopedic Procedure p=0.266, Transfusion p=0.926, Mechanical Ventilation Days p=0.957, Vascular Access Insertion p=0.889, and Drainage Intervention p=0.563.
The findings depend on meeting Chi-square assumptions, including adequate sample sizes and expected cell frequencies of at least 5. While sparse data may affect reliability, the Chi-square test remains a robust tool for analyzing categorical relationships. These results highlight the importance of NSGY Intervention and Intubation in trauma care, warranting further study to confirm these associations and explore their implications.
Line # 372-377: Findings based on table 6 and 7 has been explained in result and discussion section of our manuscript.
We have also included it in our method section.
Line # 136-139: The relationships between the MOI and medical procedures were analyzed using the Chi-square test, a suitable method for evaluating associations between categorical variables like the mechanism of injury and binary medical procedure outcomes. The test identifies whether the observed distribution deviates significantly from chance.
We hope you find our manuscript suitable for publication and look forward to hearing from you.
Sincerely,
Bharti Sharma (Corresponding author)

Round 3
Reviewer 4 Report
Comments and Suggestions for Authors
Thank you for Your detailed answers. The work has progressed significantly; however, several additional clarifications still needed:
1. Please include in your analysis the risk factors discussed in the literature as directly related to the severity of trauma, such as the height from which the fall occurred (directly related to the age of the patient - concerning more than three times the body height - 5m for an adult and than 3m for a child), the surface on which the fall occurred, high-speed impact (over 10km/h), high velocity, large caliber, or high penetrability of the projectile in the case of gunshot wounds, penetrating trauma by bladed weapons.
2. Please comment on how this interpretation of trauma kinematics can provide clarification regarding lesion typology, severity of lesions, their evolution, and the.
3. Please also discuss how this perspective on interpreting traumatic profiles could impact the proactive management of paramedical teams operating at the accident scene, in the suspected presence of complex trauma profiles whose symptoms are not immediately apparent but have significant severity and an aggravating evolutionary potential.
Author Response
Dear reviewer,
Thank you for providing these comments. We have carefully considered each comment to respond as effectively as possible. Please see our point-by-point response below
Reviewer 1 (Can be improved)
- Please include in your analysis the risk factors discussed in the literature as directly related to the severity of trauma, such as the height from which the fall occurred (directly related to the age of the patient - concerning more than three times the body height - 5m for an adult and than 3m for a child), the surface on which the fall occurred, high-speed impact (over 10km/h), high velocity, large caliber, or high penetrability of the projectile in the case of gunshot wounds, penetrating trauma by bladed weapons.
Response: At present, we have 7 tables and 1 figure. To respect the journal’s guidelines, we cannot introduce any new analytical table or figure but we have introduced all relevant findings within the manuscript. But, as per the reviewer’s suggestions, we have incorporated the following explanations:
Lines # 452-460: Falls from heights are associated with high-energy impacts and are a major cause of morbidity and mortality. Although the reliability of height as an indicator of severity has been questioned, it remains one of the most commonly used parameters in trauma assessment. Research indicates a direct relationship between the height of the fall and the likelihood of death (26,27). In SRI’s, falls are prevalent. Demographics and risk factors have been documented in observational studies and cohorts, with proposed measures to reduce the frequency and morbidity of these accidents [28]. A significant correlation exists between free fall height and injury severity. Although it is not linear, greater fall heights correspond to higher Injury Severity Scores (ISS) (29)
Lines # 171-178: Age is reported as the mean and standard deviation, while gender, race, and ethnicity are displayed as counts and percentages. The data show that individuals involved in falls had an average age of 52.51 years, the oldest among the groups, while those in the "other" injury category had the youngest average age (33.62 years). Table 2 of this paper presents the demographic breakdown of subway-related trauma cases based on different mechanisms of injury (MOI), including falls, assaults, train strikes, suicides, and other causes (such as electric injuries, strikes against objects, and other unspecified injuries).
Lines # 171-178: Table 3 of this paper outlines the injury patterns associated with various MOI in subway-related trauma incidents, specifically focusing on assaults, falls, other causes (such as electric injuries and being struck against objects), suicides, and train strikes. The table provides counts and percentages for different types of injuries, including brain-specific injuries, fractures, solid organ injuries, traumatic thoracic injuries, abdominal injuries, and traumatic amputations.
We have included all relevant information related to severity such as Trauma type (Blunt and penetrating), GCS, AIS, ISS, and numbers accounted for falls along with other mechanisms of injuries. Notably, falls accounted for the highest percentage of brain-specific injuries (65.60%) and fractures (67.50%), indicating their prevalence in this MOI. In our patient population, we found fractures of long bone, hand, foot, and face. Assaults showed a significant occurrence of traumatic thoracic injuries, with 28.90% of cases reporting these injuries. Falls accounted for the highest percentage of Solid organ injury (65.60%). Lines (281-283) Our results also show that timely and effective trauma care likely contributed to the favorable outcomes observed in many patients, with a large proportion of fall patients (72.5%) being discharged home or to a shelter.
While this study provides valuable insights into the injury patterns and clinical outcomes resulting from subway-related accidents, several limitations must be acknowledged. First, as a single-center retrospective study, the findings may not be generalizable to other trauma centers or urban settings. The patient population treated at one trauma center may not capture the full spectrum of SRIs across different regions or hospitals. This limits the broader applicability of the results.
Also, Data was collected from the NTRACS Trauma Registry, which may not always provide complete or detailed information. This may limit the generalizability of the findings to other settings with different demographic and socioeconomic characteristics. A multicenter study incorporating all emergency medical services’ data would provide a more accurate description of the injuries. These limitations suggest that further research, incorporating more comprehensive and geographically diverse data, is necessary to fully understand the epidemiology of SRIs in urban settings.
A limitation of this study is the potential misinterpretation from recoding categorical variables, especially for outcomes like discharge disposition and injury patterns. When mutually exclusive outcomes are recoded into binary indicators, unintended correlations may appear, as both categories can be flagged as “positive,” inflating associations.
Our study's reliance on ICD injury descriptions and AIS body region classifications introduces the potential for bias. Differences in coding practices and documentation quality across clinicians can affect data accuracy, making it difficult to standardize injury descriptions or severity levels across all cases. Moreover, the retrospective nature of the study prevents the establishment of causal relationships between injury mechanisms and clinical outcomes, as it only captures information available in medical records.
Another limitation is the possibility of underreporting of cases, particularly for patients with less severe injuries who may not seek immediate medical attention or those whose injuries were not documented due to inconsistent reporting practices. Therefore, the true incidence of SRIs is likely underestimated.
- Please comment on how this interpretation of trauma kinematics can provide clarification regarding lesion typology, severity of lesions, their evolution, and the..
Response: It Seems like this comment is incomplete but based on our understanding of this comment, please see the following response:
Our Hospital (Elmhurst Hospital Center) trauma registry utilizes NTRACS software (American College of Surgeons National Trauma Registry System). We requested data from the Trauma registry at EHC including PHI and PHIs were used for patient chart review (wherever applicable). Based on these 2 available sources we have used AIS, GCS, and ISS associated with all injury patterns exploring their occurrences, and severity (AIS (by parts), ISS and ISS overall, GCS), and other relevant data to support our Primary and Secondary Objective. Our primary objective is to study various traumatic injury patterns and mechanisms of injury associated with subway traumatic incidents whereas our secondary objective is to study various clinical outcomes/discharge dispositions associated with these injury patterns.
- Please also discuss how this perspective on interpreting traumatic profiles could impact the proactive management of paramedical teams operating at the accident scene, in the suspected presence of complex trauma profiles whose symptoms are not immediately apparent but have significant severity and an aggravating evolutionary potential.
Response: As per the reviewer’s suggestions, we have incorporated the following explanations:
Lines 426-438: Paramedics play an essential role in patient care. Quality of care provided by medical teams, and paramedical/EMT can be affected by the patient’s critical condition or severity of injury. For example, prehospital intubation can result in higher mortality than in patients who are non-prehospital intubated. Interventions such as intubation might result in deleterious outcomes [14]. Additionally, a more recent study reviewed subway-related injury cases, revealing a mortality rate of 9.6% and that 25% of patients suffered extremity amputations, highlighting the severe nature of these injuries [15-17]. Of the survivors, 35.4% required transfer to psychiatric or rehabilitation services, with the remaining 53.4% discharged, aligning with the trends observed in our study regarding the need for extensive rehabilitation and psychiatric care [14]. This data further emphasizes the importance of addressing long-term outcomes in subway-related trauma, including psychological and rehabilitative care, as many survivors face ongoing physical and cognitive challenges.
Lines 396-399: ANOVA revealed a statistically significant effect of MOI on RTS indicating that trauma severity varies significantly across the different MOI. We found significant associations for NSGY Intervention and Intubation indicating their likelihood is related to the mech-anism of injury.
Lines 197-201: A correlation matrix of injury patterns and hospital dispositions is provided. “…the correlation matrix highlights expected clinical patterns, such as the relationship between discharge destinations and patient stability, while also revealing the importance of how data is coded, especially for mutually exclusive outcomes like discharge to home and rehab. This correlation analysis allows clinicians to better anticipate patient needs post-discharge and make informed decisions about resource allocation.”
(Lines 174-177) The Phi coefficient correlation matrix provides valuable insights into associations between various binary clinical outcomes, informing patient care planning and resource allocation. Although discharge to home and discharge to subacute rehab are theoretically mutually exclusive options. Trauma patients were usually discharged to either one or the other, not both.
(Lines 187-196) There was a moderate positive correlation between amputation and discharge to hospice care (0.27), indicating that patients who undergo amputation may often be dealing with severe or end-of-life conditions and poor prognosis, underscoring the importance of hospice care planning for these patients. Negative associations were also notable, particularly between discharge to home and both mortality (-0.42). These associations imply that patients discharged to home are generally in more stable conditions. Many other variable pairs showed weak or minimal correlations, such as solid organ injury patterns and psychiatric care discharge disposition outcomes, indicating that these factors may operate independently in this dataset.
(Lines 281-283) Our results also show that timely and effective trauma care likely contributed to the favorable outcomes observed in many patients, with a large proportion of fall patients (72.5%) being discharged home or to a shelter.
(lines 351-362) Our findings indicate that citywide efforts to prevent subway-related trauma should be explored further. Some strategies for preventing subway injuries include enhancing surveillance by station staff, job positions for staff to take care of elderly or disabled people, education programs, training, media intervention, limiting public access to subway tracks, and optimizing platform design. These measures can effectively reduce the incidence of injuries and help manage media coverage of subway suicides (14). Unemployment is associated with a 2- to 3-fold increase in the relative risk of suicide. These studies could also assist in determining a specific unemployment rate that warrants heightened awareness and monitoring. Additionally, another effective strategy to reduce the occurrence of these tragic outcomes is to identify and target populations that are at higher risk (14). Interventions targeting at-risk populations identified in this study may decrease the morbidity and mortality associated with subway-related trauma (11).
Data was collected from the NTRACS Trauma Registry, which may not always provide complete or detailed information. This may limit the generalizability of the findings to other settings with different demographic and socioeconomic characteristics. A multicenter study incorporating all emergency medical services’ data would provide a more accurate description of the injuries. These limitations suggest that further research, incorporating more comprehensive and geographically diverse data, is necessary to fully understand the epidemiology of SRIs in urban settings.
Finally, although our study provides valuable data on immediate clinical outcomes, it does not address the long-term consequences or complications of these injuries. Understanding the long-term impact on survivors’ quality of life, cognitive functioning, and healthcare utilization would offer a more comprehensive picture of subway-related trauma.
We hope our responses meet your expectations and we look forward to your feedback.
Sincerely,
Bharti Sharma (Corresponding author)
